# Adam with model exponential moving average is effective for nonconvex optimization

**Kwangjun Ahn**
Microsoft Research
Cambridge, MA
kwangjunahn@microsoft.com

**Ashok Cutkosky**
Boston University
Boston, MA
ashok@cutkosky.com

## Abstract

In this work, we offer a theoretical analysis of two modern optimization techniques for training large and complex models: (i) adaptive optimization algorithms, such as Adam, and (ii) the model exponential moving average (EMA). Specifically, we demonstrate that a clipped version of Adam with model EMA achieves the optimal convergence rates in various nonconvex optimization settings, both smooth and nonsmooth. Moreover, when the scale varies significantly across different coordinates, we demonstrate that the coordinate-wise adaptivity of Adam is provably advantageous. Notably, unlike previous analyses of Adam, our analysis crucially relies on its core elements—momentum and discounting factors—as well as model EMA, motivating their wide applications in practice.

## 1 Introduction

In neural network training, the training loss $F : \mathbb{R}^d \to \mathbb{R}$ is often optimized using an iterative optimization algorithm which starts with the initial iterate $\mathbf{x}_0$ and then updates during each iteration $t = 1, 2, \ldots$ as follows:

$$\mathbf{x}_t = \mathbf{x}_{t-1} + \mathbf{z}_t \,,$$

where $\mathbf{z}_t$ denotes the increment chosen by the algorithm during the $t$-th iteration. One of the most popular optimization algorithms is **Adam** [Kingma and Ba, 2014]. Adam has gained significant attention due to its effectiveness in training Transformer-based language models [Zhang et al., 2020a, Kunstner et al., 2023, Jiang et al., 2023, Pan and Li, 2023, Ahn et al., 2024a, Kunstner et al., 2024, Zhang et al., 2024a].

The **model exponential moving average** (EMA) [Polyak and Juditsky, 1992, Ruppert, 1988] is an optimization technique that has gained popularity in conjunction with Adam for various recent applications. EMA maintains an exponential moving average of the model iterates, $\mathbf{x}_t$, which contributes to the stabilization of these iterates. There has been a resurgence of interest in this technique due to its effectiveness in training high-quality generative models [Yaz et al., 2018, Karras et al., 2019, Song et al., 2021b, Dhariwal and Nichol, 2021, Nichol and Dhariwal, 2021, Song et al., 2021a, Balaji et al., 2022, Karras et al., 2022, Rombach et al., 2022, Kang et al., 2023, Karras et al., 2023]. Moreover, a recent work by Block et al. [2024] demonstrates the effectiveness of EMA for both language modeling and imitation learning applications.

In this work, we theoretically study the effectiveness of these two modern optimization techniques. Our main results can be informally summarized as follows.

> **Theorem 1** (Informal). CLIPPED-ADAM *with the EMA on its iterates achieves the optimal convergence rate for nonconvex optimization both for smooth and nonsmooth settings (Section 5). The coordinate-wise adaptivity of Adam is particularly effective when the scale varies across different coordinates (Section 6).*

38th Conference on Neural Information Processing Systems (NeurIPS 2024).

Our main results are based on the online-to-nonconvex conversion framework of Cutkosky et al. [2023], which chooses the increment $\mathbf{z}_t$ based on an online learner of choice. In particular, our approach is quite different than the previous analyses of Adam (see Section 1.1 below). Notably, our analysis relies on the key components of Adam (momentum and adaptive learning rate) as well as EMA of the iterates, offering new, theoretical insight into their success. See Section 7 for a more detailed discussion.

At a high level, our analysis combines the main insights from the two recent works: Zhang and Cutkosky [2024] and Ahn et al. [2024b]. We first carefully modify the discounted-to-nonconvex conversion framework (Lemma 7) of Zhang and Cutkosky [2024] which converts an online learner that achieves a good discounted regret (Definition 6) into a good noncovex optimizer. We then combine it with the main insight of Ahn et al. [2024b] that an effective discounted online learner can be designed based on scale-free Follow-the-Regularized-Leader (FTRL) [Orabona and Pál, 2018]. In particular, the way we arrive at Adam is similar to Ahn et al. [2024b]: choosing a discounted version of FTRL in the discounted-to-nonconvex conversion leads to Adam.

## 1.1 Related work

Even though Adam is widely used in deep learning, our theoretical understanding of its inner workings, especially the importance of its core components—momentum and discounting factors—remains incomplete, as pointed out by Ahn et al. [2024b]. Most theoretical work on Adam and its variations focus on characterizing the convergence rate for convex or smooth nonconvex functions, where methods like SGD already achieve the minimax optimal convergence rate. [Reddi et al., 2018, Zhou et al., 2019, Chen et al., 2019, Zou et al., 2019, Alacaoglu et al., 2020, Guo et al., 2021, Défossez et al., 2022, Zhang et al., 2022, Li et al., 2023, Wang et al., 2023]. In fact, even the most recent results [Li et al., 2023, Wang et al., 2023] are not reflective of practice, in the sense that Adam's convergence rate worsens with momentum [Wang et al., 2023, §6] or is no better than that of SGD [Li et al., 2023, §7]. A notable exception is Crawshaw et al. [2022], which demonstrates the advantages of momentum under the generalized smoothness conditions of Zhang et al. [2020a]. However, the algorithm they analyze is signSGD, which differs significantly from the original Adam. In contrast, as we will explore in the subsequent sections, momentum and discounting factors are important in our analysis. See Section 7 for more details.

We also highlight that our analysis relies on model EMA, a technique widely used in practice as mentioned above (also see a recent work by Block et al. [2024]). It is worth noting that EMA (or model averaging in general) has shown to have generalization benefits in practice [Tarvainen and Valpola, 2017, Izmailov et al., 2018]. In this paper, we study EMA from an optimization perspective, and show that the use of EMA leads to optimal guarantees for nonconvex optimization. Interestingly, EMA naturally derives from the discounted-to-online conversion (see Algorithm 1), which, we believe, provides new theoretical insights into this practical method.

The use of EMA also represents a significant departure from most non-convex optimization analyses. While EMA is a classical technique in the *convex* setting, theoretical analyses in the non-convex setting typically randomly select an iterate as the "final output" of the optimizer, rather than using EMA. This random selection is intuitively extremely impractical (indeed, on average it actually wastes half of the computation), and is never performed in real implementations.

Our analysis follows a line of work studying convergence guarantees for non-smooth non-convex optimization. Our particular convergence criterion is similar to finding the Goldstein stationary points [Goldstein, 1977] that were first studied in the context of modern machine learning by [Zhang et al., 2020b], and has seen much subsequent interest [Tian and So, 2022, Jordan et al., 2023, Davis et al., 2020]. Other notions of convergence are also reasonable—common alternatives involve the Moreau envelope, or imposing a weak convexity condition [Davis et al., 2018, 2022a].

## 2 Setting for nonconvex and nonsmooth optimization

Throughout this paper, unless specified otherwise, $\|\cdot\|$ denotes the $L_2$ norm. Following [Cutkosky et al., 2023], we consider optimizing a loss function $F$ that satisfies the following conditions, accessing information about $F$ through a *stochastic gradient oracle* $\textsc{StoGrad} : \mathbb{R}^d \times \mathcal{Z} \to \mathbb{R}^d$, for the set of randomness $\mathcal{Z}$.

**Assumption 2.** *Let $F : \mathbb{R}^d \to \mathbb{R}$ be a differentiable function with the following properties:*

- *Let $\Delta := F(\mathbf{x}_0) - \inf_{\mathbf{x}} F(\mathbf{x})$.*

- *For any two points $\mathbf{x}$ and $\mathbf{y}$, $F(\mathbf{y}) - F(\mathbf{x}) = \int_0^1 \langle \nabla F(\mathbf{x} + t(\mathbf{y} - \mathbf{x})), \mathbf{y} - \mathbf{x} \rangle \, dt$.*

- **Lipschitzness.** *$F$ is $G$-Lipshitz, i.e., for any point $\mathbf{x}$, $\|\nabla F(\mathbf{x})\| \leq G$.*

- **Stochastic gradient variance.** *For any point $\mathbf{x}$, the stochastic gradient $\mathbf{g} \leftarrow \text{STOGRAD}(\mathbf{x}, r)$ for randomness $r \in \mathcal{Z}$ satisfies $\mathbb{E}[\mathbf{g}] = \nabla F(\mathbf{x})$ and $\mathbb{E}\|\mathbf{g} - \nabla F(\mathbf{x})\|^2 \leq \sigma^2$.*

The Lipschitz continuity condition is a standard assumption in nonconvex nonsmooth settings. However, as we will discuss in Section 6, one of the key insights from our results is that Adam enables us to **adapt** to the Lipschitz constants **coordinate-wise** without requiring prior knowledge of these constants. Note that we almost certainly need some form of structural assumption on the "difficulty" of the loss function; thus, relaxing the Lipschitz assumption would likely come at the cost of another assumption, such as smoothness.

The second condition, called *well-behavedess* in [Cutkosky et al., 2023, Definition 1], is a mild regularity condition. For any locally Lipschitz function $F$, applying an arbitrarily small perturbation to the function is sufficient to ensure this condition [Cutkosky et al., 2023, Proposition 2].

For the notion of optimality, we follow Zhang and Cutkosky [2024] and consider the following notion of stationarity for nonconvex and nonsmooth functions. This notion is a slight relaxation of the notion of a Goldstein stationarity point [Goldstein, 1977], which was further studied by recent works [Zhang et al., 2020a, Davis et al., 2022b, Tian et al., 2022, Jordan et al., 2023].

**Definition 3 ($(\lambda, \varepsilon)$-stationary point).** *Suppose $F : \mathbb{R}^d \to \mathbb{R}$ is differentiable. We say $\mathbf{x}$ is a $(\lambda, \varepsilon)$-stationary point of $F$ if $\|\nabla F(\mathbf{x})\|^{[\lambda]} \leq \varepsilon$, where*

$$\|\nabla F(\mathbf{x})\|^{[\lambda]} := \inf_{\substack{p \in \mathcal{P}(\mathbb{R}^d), \\ \mathbb{E}_{\mathbf{y} \sim p}[\mathbf{y}] = \mathbf{x}}} \left\{ \|\mathbb{E}[\nabla F(\mathbf{y})]\| + \lambda \cdot \mathbb{E}\|\mathbf{y} - \mathbf{x}\|^2 \right\} .$$

To further motivate this definition, we remark that $(\lambda, \epsilon)$-stationary points retain the desirable properties of Goldstein stationary points. Specifically, the following result [Zhang and Cutkosky, 2024, Lemma 2.3] demonstrates that, akin to Goldstein stationary points, $(\lambda, \epsilon)$-stationary points can be reduced to first-order stationary points with appropriate choices of $\lambda$ when the objective function is smooth or second-order smooth.

**Lemma 4.** *If $F$ is $L$-smooth, then an $(L^2\varepsilon^{-1}, \varepsilon)$-stationary point $\mathbf{x}$ of $F$ satisfies $\|\nabla F(\mathbf{x})\| \leq 2\varepsilon$. Moreover, if $F$ is $H$-second-order-smooth, then an $(H/2, \varepsilon)$-stationary point $\mathbf{x}$ of $F$ satisfies $\|\nabla F(\mathbf{x})\| \leq 2\varepsilon$.*

Moreover, as shown by [Zhang and Cutkosky, 2024, Lemma 2.4], $(\lambda, \varepsilon)$-stationary points can also be reduced to Goldstein stationary points when $F$ is Lipschitz.

**Lemma 5.** *Suppose $F$ is $G$-Lipschitz. For any $\lambda, \varepsilon, \delta > 0$, a $(\lambda, \varepsilon)$-stationary point is a $(\delta, \varepsilon')$-Goldstein stationary point, where $\varepsilon' = (1 + \frac{2G}{\lambda\delta^2}) \cdot \varepsilon$.*

Now we design algorithms that find $(\lambda, \varepsilon)$-stationary points efficiently.

## 3 Discounted-to-nonconvex conversion: online learning of increments

Our main results are built on the online-to-nonconvex conversion framework of Cutkosky et al. [2023]. At its core, this framework involves selecting the increment $\mathbf{z}_t$ using an online learner, as discussed by Ahn et al. [2024b]. Specifically, we follow a variant developed by Zhang and Cutkosky [2024], which carefully incorporates the discounting factor in the conversion process. Note that we make slight modifications to the version proposed by Zhang and Cutkosky [2024] as follows. Here $\text{Exp}(1)$ denotes the exponential random variable with mean 1.

Given Algorithm 1, it turns out we need to design an online learner that minimizes the discounted regret, formally defined below. It is worth noting that discounted regret has been recently studied with the goal of better adapting online learners to dynamic environments [Ahn et al., 2024b, Zhang et al., 2024b, Jacobsen and Cutkosky, 2024].

---

**Algorithm 1 Discounted-to-nonconvex conversion** (choosing increments via online learning)

---

**Input:** Initial point $\mathbf{x}_0$, $T \in \mathbb{N}$, online learning algorithm $\mathcal{A}$, and discounting factor $\beta \in (0,1)$
**for** $t = 1, 2 \ldots, T$ **do**
    Receive $\mathbf{z}_t$ from $\mathcal{A}$ // choose the increment using an online learner
    Update $\mathbf{x}_t \leftarrow \mathbf{x}_{t-1} + \alpha_t \mathbf{z}_t$, where $\alpha_t \sim \text{Exp}(1)$ *i.i.d.*
    Compute $\mathbf{g}_t \leftarrow \text{STOGRAD}(\mathbf{x}_t, r_t)$ with freshly sampled randomness $r_t$
    Send $\ell_t^{[\beta]}(\mathbf{z}) := \langle \beta^{-t} \mathbf{g}_t, \mathbf{z} \rangle$ to $\mathcal{A}$
    // Maintain exponential moving average (for output only):
    Update $\overline{\mathbf{x}}_t \leftarrow \frac{\beta - \beta^t}{1 - \beta^t} \overline{\mathbf{x}}_{t-1} + \frac{1-\beta}{1-\beta^t} \mathbf{x}_t$     (Equivalently, $\overline{\mathbf{x}}_t \leftarrow \frac{1-\beta}{1-\beta^t} \sum_{s=1}^{t} \beta^{t-s} \mathbf{x}_s$)
**end for**

---

**Definition 6 (Discounted regret).** *For a comparator* $\mathbf{u}$*, the* $\beta$*-discounted regret is defined as*

$$\text{Regret}_t^{[\beta]}(\mathbf{u}) := \beta^t \cdot \sum_{s=1}^{t} (\ell_s^{[\beta]}(\mathbf{z}_s) - \ell_s^{[\beta]}(\mathbf{u})) = \sum_{s=1}^{t} \beta^{t-s} \langle \mathbf{g}_s, \mathbf{z}_s - \mathbf{u} \rangle .$$

The discounted regret of an online learner $\mathcal{A}$ can be used to upper bound the norm of averaged gradients, as shown in the following result.

---

**Lemma 7 (Discounted-to-nonconvex conversion).** *Suppose that $F$ satisfies Assumption 2. Then for the comparator sequence chosen as* $\mathbf{u}_t := -D \frac{\sum_{s=1}^{t} \beta^{-s} \nabla F(\mathbf{x}_s)}{\left\| \sum_{s=1}^{t} \beta^{-s} \nabla F(\mathbf{x}_s) \right\|}$*, Algorithm 1 gives*

$$\mathbb{E}_{t \sim [T]} \mathbb{E} \left\| \mathbb{E}_{\mathbf{y}_t} \nabla F(\mathbf{y}_t) \right\| \leq \frac{\Delta}{DT} + \frac{2G + \sigma}{(1-\beta)T} + \sigma \sqrt{1-\beta}$$
$$+ \frac{1}{DT} \left[ \beta \cdot \mathbb{E} \left[ \text{Regret}_T^{[\beta]}(\mathbf{u}_T) \right] + (1-\beta) \cdot \sum_{t=1}^{T} \mathbb{E} \left[ \text{Regret}_t^{[\beta]}(\mathbf{u}_t) \right] \right],$$

*where $\mathbf{y}_t$ is distributed over $\{\mathbf{x}_s\}_{s=1}^{t}$ as* $\mathbb{P}(\mathbf{y}_t = \mathbf{x}_s) = \beta^{t-s} \cdot \frac{1-\beta}{1-\beta^t}$ *for $s = 1, 2, \ldots, t$.*

---

The proof combines the techniques of [Cutkosky et al., 2023, Theorem 7] and [Zhang and Cutkosky, 2024, Theorem 3.3]. See Appendix A for details.

We briefly explain how Lemma 7 can be used to find a $(\lambda, \epsilon)$-stationary point (Definition 3). Recall that $(\lambda, \epsilon)$-stationarity essentially requires producing a point $\mathbf{x} = \mathbb{E}[\mathbf{y}]$ such that both $\|\mathbb{E}[\nabla F(\mathbf{y})]\|$ and $\mathbb{E}\|\mathbf{y} - \mathbf{x}\|^2$ are small.

Given this context, Lemma 7 states that as long as the discounted regret of the online learner $\mathcal{A}$ is small, we can ensure that the EMA iterates $\overline{\mathbf{x}}_t = \mathbb{E}[\mathbf{y}_t]$ serve as good candidates for $(\lambda, \epsilon)$-stationarity, since the term $\mathbb{E} \left\| \mathbb{E}_{\mathbf{y}_t} \nabla F(\mathbf{y}_t) \right\|$ can be kept small. The remaining task is to bound the variance term, $\mathbb{E} \left\| \mathbf{y}_t - \overline{\mathbf{x}}_t \right\|^2$, which will be addressed later in Lemma 10.

Moreover, the comparator $\mathbf{u}_t$ roughly models the *update direction that an oracle algorithm with perfect knowledge of the loss would select*. In the proof of Lemma 7, we demonstrate that moving along the $\mathbf{u}_t$ direction effectively decreases the loss value, which forms the basis for establishing our convergence guarantee.

Thanks to the discounted-to-nonconvex conversion, the task now reduces to designing an online learner that achieves low discounted regret.

## 4   Scale-free Follow-the-Regularized-Leader (FTRL)

In this section, we introduce an algorithmic component, called the Followed-The-Regularized-Leader (FTRL), a powerful online learning technique with various applications [Gordon, 1999, Kalai and Vempala, 2005, Shalev-Shwartz and Singer, 2006, Abernethy et al., 2008, Nesterov, 2009, Hazan and Kale, 2010].

For the setting, consider the online linear optimization (OLO) setting, where during each round $t = 1, \ldots, T$, and online learner chooses $\mathbf{z}_t$, and then the linear loss $\ell_t(\cdot) = \langle \mathbf{v}_t, \cdot \rangle$ is revealed by the environment. Here the goal of the online learner is to minimize the regret defined as $\sum_t \langle \mathbf{v}_t, \mathbf{z}_t - \mathbf{u} \rangle$, where $\mathbf{u}$ is the comparator in hindsight. For this setting, FTRL is presented in Algorithm 2.

---

**Algorithm 2** Follow-the-Regularized-Leader (FTRL)

---

**Require:** Regularizers $\{\psi_t\} : \mathbb{R}^d \to \mathbb{R}$, the domain $\mathcal{D} \subseteq \mathbb{R}^d$
1: **for** $t = 1, 2, \ldots, T$ **do**
2:      Update $\mathbf{z}_t \leftarrow \operatorname{argmin}_{\mathbf{z} \in \mathcal{D}} \left[ \psi_t(\mathbf{z}) + \sum_{s=1}^{t-1} \ell_s(\mathbf{z}) \right]$
3:      Receive the next loss $\ell_t(\cdot) = \langle \mathbf{v}_t, \cdot \rangle$
4: **end for**

---

The key insight of Ahn et al. [2024b] and Zhang et al. [2024b] is that in order to design an online learner for discounted regret, it is important that the online learner is *scale-free* as described below. In particular, following Ahn et al. [2024b], we consider a gradient adaptive scale-free FTRL algorithm called *scale-free FTRL* [Orabona and Pál, 2018].

We will focus on the case where $\mathcal{D} = \mathbb{B}_D$, the $d$-dimensional $L_2$-ball of radius $D > 0$. Scale-free FTRL is given by Algorithm 2 with the following choie:

$$\psi_t(\cdot) = \frac{1}{\eta_t} \left\| \cdot \right\|^2 \quad \text{and} \quad \eta_t = \frac{D}{\sqrt{\sum_{s=1}^{t-1} \left\| \mathbf{v}_s \right\|^2}}.$$

Then using the clipping operator $\operatorname{clip}_D(\mathbf{x}) := \mathbf{x} \min(D/\|\mathbf{x}\|, 1)$, we can write down the update rule more explicitly as follows:

$$\mathbf{z}_t \leftarrow \operatorname*{argmin}_{\mathbf{z} \in \mathbb{B}_D} \left[ \frac{1}{\eta_t} \left\| \mathbf{z} \right\|^2 + \sum_{s=1}^{t-1} \langle \mathbf{v}_s, \mathbf{z} \rangle \right] = -\operatorname{clip}_D \left( D \frac{\sum_{s=1}^{t-1} \mathbf{v}_s}{\sqrt{\sum_{s=1}^{t-1} \left\| \mathbf{v}_s \right\|^2}} \right). \quad \text{(SCALE-FREE FTRL)}$$

Here, if the denominator is zero, *i.e.*, $\mathbf{v}_1 = \cdots = \mathbf{v}_{t-1} = \mathbf{0}$, then we set $\mathbf{z}_t \leftarrow 0$. Note that this algorithm is scale-free in the sense that when the loss sequence is scaled by a scalar $c > 0$, the updates remain the same.

Let us now present the regret bound of SCALE-FREE FTRL.

**Lemma 8** (Gradient-adaptive regret bound). *For any $T > 0$, loss sequence $\mathbf{v}_{1:T}$ and comparator $\mathbf{u} \in \mathbb{R}^d$ s.t. $\|\mathbf{u}\| \leq D$, SCALE-FREE FTRL guarantees the following regret bound:*

$$\sum_{t=1}^{T} \langle \mathbf{v}_t, \mathbf{z}_t - \mathbf{u} \rangle \leq 4D \sqrt{\sum_{t=1}^{T} \left\| \mathbf{v}_t \right\|^2}.$$

We note that Lemma 8 follows (with a slightly worse constant) from [Orabona and Pál, 2018, Theorem 1], and the version we invoke is here due to [Ahn et al., 2024b, Theorem A.1].

Recall from Lemma 7 that an online learner for the discounted-to-nonconvex conversion (Algorithm 1) needs to have a low discounted regret. To achieve this, following Ahn et al. [2024b] and Zhang et al. [2024b], we simply substitute $\mathbf{v}_t \leftarrow \beta^{-t} \mathbf{g}_t$ into SCALE-FREE FTRL, resulting in the update

$$\mathbf{z}_t \leftarrow -\operatorname{clip}_D \left( D \frac{\sum_{s=1}^{t-1} \beta^{-s} \mathbf{g}_s}{\sqrt{\sum_{s=1}^{t-1} \beta^{-2s} \left\| \mathbf{g}_s \right\|^2}} \right). \quad (\beta\text{-FTRL})$$

Here again, if the denominator is zero, *i.e.*, $\mathbf{g}_1 = \cdots = \mathbf{g}_{t-1} = \mathbf{0}$, then we set $\mathbf{z}_t \leftarrow 0$. Then, the following result characterizes the discounted regret guarantee of $\beta$-FTRL.

**Theorem 9** (**Discounted regret bound**). *Let $\beta \in (0, 1]$. For any $T > 0$, loss sequence $\mathbf{g}_{1:T}$ and comparator $\mathbf{u} \in \mathbb{R}^d$ s.t. $\|\mathbf{u}\| \leq D$, $\beta$-FTRL guarantees the following static regret bound*

$$\operatorname{Regret}_T^{[\beta]}(\mathbf{u}) \leq 4D \sqrt{\sum_{t=1}^{T} \beta^{2(T-t)} \left\| \mathbf{g}_t \right\|^2}.$$

We next use this result to design an algorithm for nonconvex optimization.

# 5 Discounted-FTRL leads to adaptive nonconvex optimization

In this section, as a warm-up, let us see the implications of choosing $\mathcal{A} = \beta$-FTRL in Algorithm 1. First, let us obtain a bound on the expected discounted regret. By Theorem 9 together with Jensen's inequality, we have the following regret bound for any $t = 1, 2, \ldots, T$:

$$\mathbb{E}\left[\text{Regret}_t^{[\beta]}(\mathbf{u}_t)\right] \leq 4D \, \mathbb{E} \sqrt{\sum_{t=1}^{T} \beta^{2(T-t)} \|\mathbf{g}_t\|^2} \leq 4D \sqrt{\sum_{t=1}^{T} \beta^{2(T-t)} \, \mathbb{E} \|\mathbf{g}_t\|^2} \, .$$

Since $\mathbb{E} \|\mathbf{g}_t\|^2 \leq G^2 + \sigma^2$ and $\frac{1}{\sqrt{1-\beta^2}} \leq \frac{1}{\sqrt{1-\beta}}$, it follows that

$$\mathbb{E}\left[\text{Regret}_t^{[\beta]}(\mathbf{u}_t)\right] \leq \frac{4D(G+\sigma)}{\sqrt{1-\beta^2}} \leq \frac{4D(G+\sigma)}{\sqrt{1-\beta}} \, . \tag{1}$$

## 5.1 From gradient-adaptive regret to nonconvex optimization

In order to obtain nonconvex optimization guarantees in terms of the $(\lambda, \varepsilon)$-stationarity (Definition 3), we need to handle the variance term. Following [Zhang and Cutkosky, 2024, Lemma 3.2], the variance term can be bounded as follows.

**Lemma 10** (Variance bound). *Using the notations of Lemma 7, for any $t = 1, 2, \ldots, T$, $\beta$-FTRL satisfies*

$$\mathbb{E}_{t\sim[T]} \mathbb{E} \|\mathbf{y}_t - \overline{\mathbf{x}}_t\|^2 \leq 12 \frac{D^2}{(1-\beta)^2} \, .$$

*Proof.* From [Zhang and Cutkosky, 2024, Lemma 3.2], it follows that $\mathbb{E} \sum_{t=1}^{T} \|\mathbf{y}_t - \overline{\mathbf{x}}_t\|^2 \leq \frac{12}{(1-\beta)^2} \mathbb{E} \sum_{t=1}^{T} \|\mathbf{z}_t\|^2$. Now since $\|\mathbf{z}_t\| \leq D$ for all $t = 1, 2, \ldots, T$, after dividing each side by $T$, we get the desired inequality. $\square$

Plugging the regret bound (1) into Lemma 7 and combining it with Lemma 10, we arrive at the following optimization guarantee in terms of the $(\lambda, \varepsilon)$-stationarity. See Section B.1 for a proof.

---

**Theorem 11.** *Suppose that $F$ satisfies Assumption 2 and consider any $\lambda > 0$. For $C > 0$, choose $\mathcal{A} = \beta$-FTRL in Algorithm 1 with the following parameters:*

$$\beta = 1 - \left(\frac{\varepsilon}{10C}\right)^2, \quad D = \frac{(1-\beta)\varepsilon^{1/2}}{4\lambda^{1/2}}, \quad \text{and} \quad T = \frac{1}{1-\beta} \cdot \max\left\{\frac{4\Delta\lambda^{1/2}}{\varepsilon^{3/2}}, \frac{12C}{\varepsilon}\right\} \, .$$

*Then we have $\mathbb{E}_{t\sim[T]} \mathbb{E} \|\nabla F(\overline{\mathbf{x}}_t)\|^{[\lambda]} \leq (1 + \frac{G+\sigma}{C})\varepsilon$. In other words, a randomly chosen **model EMA** $\overline{\mathbf{x}}_t$ is a $(\lambda, (1 + \frac{G+\sigma}{C})\varepsilon)$-stationary point, in expectation.*

---

## 5.2 Optimality and gradient adaptivity

Here, we discuss several notable aspects of the guarantee provided in Theorem 11.

### 5.2.1 Optimality

As shown in [Zhang and Cutkosky, 2024, Corollary 5.1], the lower bound on the iteration complexity for finding a $(\lambda, \varepsilon)$-stationary point is $\Omega((G + \sigma)^2 \Delta \lambda^{1/2} \varepsilon^{-7/2})$, provided that $\lambda \leq \frac{G^4}{\Delta^2} \varepsilon^{-1}$. Theorem 11 implies that setting $C = G + \sigma$ achieves this optimal iteration complexity.

**Corollary 12.** *In Theorem 11, choosing $C = G + \sigma$ leads to the following iteration complexity for finding a $(\lambda, \varepsilon)$-stationary point:*

$$O\left(\max\left\{\frac{(G+\sigma)^2 \Delta \lambda^{1/2}}{\varepsilon^{7/2}}, \frac{(G+\sigma)^3}{\varepsilon^3}\right\}\right) \, .$$

*In particular, treating $G$, $\sigma$, and $\Delta$ as constants, as long as $\lambda \gtrsim \varepsilon$, this leads to the optimal complexity of $O((G + \sigma)^2 \Delta \lambda^{1/2} \varepsilon^{-7/2})$.*

In light of Lemma 4, the above optimal complexity can be converted into the optimal complexities for smooth settings.

**Corollary 13** (Smooth settings). *Corollary 12 implies the following optimal iteration complexity for smooth settings. Choosing $\lambda = O(\varepsilon^{-1})$, it implies the optimal complexity of $O(\varepsilon^{-4})$ for smooth loss functions [Arjevani et al., 2023]. Similarly, with $\lambda = O(1)$, it achieves the optimal iteration complexitiy of $O(\varepsilon^{-7/2})$ for second-order smooth loss functions [Arjevani et al., 2020].*

We next discuss the benefits of using the gradient-adaptive regret bound (Theorem 9) by considering the case where we do not have knowledge of $G, \sigma$.

### 5.2.2 Gradient adaptivity

A remarkable consequence of Theorem 11 is that, due to the gradient-adaptive regret bound of Theorem 9, the final convergence guarantee has a better dependence on $G, \sigma$ in the case when we do not have knowledge of them. For concreteness, in the following discussion, we treat $G, \sigma, \Delta$ as constants, and focus on the regime $\lambda \gtrsim \varepsilon$.

First, our Theorem 11 with $C = 1$ (since we do not know $G, \sigma$) leads to the following iteration complexity for finding a $(\lambda, \varepsilon)$-stationary point:

$$O\left((G + \sigma)^{7/2} \Delta \lambda^{1/2} \varepsilon^{-7/2}\right)$$

The price we pay for not knowing $G, \sigma$ relative to the lower bound is a multiplicative factor of $(G + \sigma)^{3/2}$. To see the benefit of this adaptive regret approach, let us consider the guarantees given by Zhang and Cutkosky [2024]. Their approach is based on choosing online gradient descent for $\mathcal{A}$ in Algorithm 1, when the learning rate is not properly tuned with the knowledge of $G$ and $\sigma$, it would lead to the following (suboptimal) discounted regret bound:

$$\mathbb{E}\left[\text{Regret}_t^{[\beta]}(\mathbf{u}_t)\right] \leq O\left(\frac{D(G + \sigma)^2}{\sqrt{1 - \beta}}\right).$$

Then, the resulting iteration complexity becomes $O(\Delta \lambda^{1/2} (\frac{\varepsilon}{(G+\sigma)^2})^{-7/2})$, which is equal to $O\left((G + \sigma)^7 \Delta \lambda^{1/2} \varepsilon^{-7/2}\right)$. This is larger than the complexity due to our adaptive approach by a multiplicative factor of $(G + \sigma)^{7/2}$.

Next, we build on the results from this section and consider a better approach to design an adaptive nonconvex optimizer.

## 6 Coordinate-wise adaptivity via (clipped-)Adam

In this section, we consider the setting where the Lipschitzness constants vary across different coordinates, which is empirically observed to be reflective of practical neural network training (see, *e.g.* [Crawshaw et al., 2022, Zhuang et al., 2022]). Formally, we consider the following setting.

**Assumption 14.** *Under the same setting as Assumption 2, we replace the last two conditions with the following coordinate-wise version:*

- *For each coordinate $i = 1, 2, \ldots, d$, there is a Lipschitzness constant $G_i > 0$ and a variance constant $\sigma_i > 0$ such that $\forall \mathbf{x}$, $|\partial_i F(\mathbf{x})| \leq G_i$ and the stochastic gradient $\mathbf{g} \leftarrow \text{STOGRAD}(\mathbf{x}, r)$ satisfies $\mathbb{E}[\mathbf{g}[i]] = \partial_i F(\mathbf{x}_i)$ and $\mathbb{E}|\mathbf{g}[i] - \partial_i F(\mathbf{x})|^2 \leq \sigma_i^2$. (Here, $\partial_i F$ denotes the partial derivative of $F$ w.r.t. the $i$-th coordinate.)*

*Let $\boldsymbol{G} := (G_1, \ldots, G_d)$ and $\boldsymbol{\sigma} := (\sigma_1, \ldots, \sigma_d)$. Then, the above condition implies the last two conditions in Assumption 2 with $G = \|\boldsymbol{G}\|_2$ and $\sigma = \|\boldsymbol{\sigma}\|_2$.*

As we mentioned before, the previous approaches [Cutkosky et al., 2023, Zhang and Cutkosky, 2024] choose the online learner $\mathcal{A}$ to be online gradient descent, and hence choosing the learning rate requires the knowledge of $G_i, \sigma_i$ for all $i$. However, for neural network training, $d$ is equal to the number of parameters in the network, so tuning them individually is computationally infeasible. We instead consider running $\beta$-FTRL **coordinate-wise** in Algorithm 1, which will automatically adapt to each coordinate. We begin with an important observation that such an approach in fact leads to a popular optimizer widely used in practice.

## 6.1 Coordinate-wise discounted FTRL corresponds to (clipped-)Adam

For notational simplicity, fix a coordinate among $i = 1, 2, \ldots, d$, and let us denote the iterate by $x_t$, the stochastic gradient by $g_t$, and the update by $z_t$. Then the resulting optimizer becomes:

$$z_{t+1} = -\text{clip}_D \left( D \frac{\sum_{s=1}^{t} \beta_1^{t-s} g_s}{\sqrt{\sum_{s=1}^{t} \beta_2^{t-s} g_s^2}} \right), \qquad \text{(CLIPPED-ADAM)}$$

where $\beta_1 = \beta$ and $\beta_2 = \beta^2$. Here, again if the denominator is zero, *i.e.*, if $g_1 = \cdots = g_t = 0$, then we set the update to be zero, *i.e.*, $z_{t+1} = 0$. Note that **this is almost exactly the Adam optimizer** [Kingma and Ba, 2014], except that now we add clipping to control the variance of the iterates relative to their EMA. Notably, CLIPPED-ADAM retains one of the most important properties of Adam: it is *scale-invariant*. The scale invariance causes the optimizer to make updates of the same magnitude on each coordinate even when the scale differs across different coordinates.

In practice, we expect that the clipping operation will effectively be a no-op. This is because, when the algorithm is converging (even if the convergence is somewhat slow), the gradients are likely to behave as approximately mean-zero random variables (due to factors such as stochastic noise, unstable training trajectories, etc.). In such cases, standard concentration inequalities imply that $\sum_{s=1}^{t} \beta^{t-s} g_s \lesssim \sqrt{\sum_{s=1}^{t} (\beta^{t-s} g_s)^2}$, and hence, the clipping has no effect.

We also remark that CLIPPED-ADAM does not consider the "bias correction" terms in the original updates of Adam [Kingma and Ba, 2014]. However, note that the bias correction terms are coordinate-independent, and they can be merged into the scalar $D$.

## 6.2 Nonconvex optimization guarantees of CLIPPED-ADAM

We next discuss the theoretical guarantees of CLIPPED-ADAM for nonconvex and nonsmooth optimizaton. Inspired by [Duchi et al., 2010, McMahan and Streeter, 2010], where the coordinate-wise online learners lead to regret bounds with respect to the $L_1$ norms of stochastic gradients, we consider the following variant of Definition 3, in the same vein as [Cutkosky et al., 2023, Section 4].

**Definition 15** $((\lambda, \varepsilon)$-$L_1$-**stationary point).** *Suppose $F : \mathbb{R}^d \to \mathbb{R}$ is differentiable. We say $\mathbf{x}$ is a $(\lambda, \varepsilon)$-$L_1$-stationary point of $F$ if $\|\nabla F(\mathbf{x})\|_1^{[\lambda]} \leq \varepsilon$, where*

$$\|\nabla F(\mathbf{x})\|_1^{[\lambda]} := \inf_{\substack{p \in \mathcal{P}(\mathbb{R}^d), \\ \mathbb{E}_{\mathbf{y} \sim p}[\mathbf{y}] = \mathbf{x}}} \left\{ \|\mathbb{E}[\nabla F(\mathbf{y})]\|_1 + \lambda \cdot \mathbb{E} \|\mathbf{y} - \mathbf{x}\|_2^2 \right\} .$$

Using the fact $\|\cdot\|_1 \leq \sqrt{d} \|\cdot\|_2$, one can connect the two notions of $(\lambda, \varepsilon)$-stationary points.

**Lemma 16.** *A $(\lambda/\sqrt{d}, \varepsilon/\sqrt{d})$-stationary point is a $(\lambda, \varepsilon)$-$L_1$-stationary point.*

In order to obtain the guarantee in terms of $L_1$-norm, we consider the coordinate-wise version of discounted-to-online conversion, in the same vein as [Cutkosky et al., 2023, Appendix G]. See Section A.1 for details.

**Lemma 17** ($L_1$-**variant of Lemma 7**). *Suppose that $F$ satisfies Assumption 14. Consider the comparator sequence chosen as $\mathbf{u}_t$ defined as $\mathbf{u}_t[i] := -D \frac{\sum_{s=1}^{t} \beta^{-s} \partial_i F(\mathbf{x}_s)}{\left| \sum_{s=1}^{t} \beta^{-s} \partial_i F(\mathbf{x}_s) \right|}$ for $i = 1, 2, \ldots, d$. Then, Algorithm 1 gives*

$$\mathbb{E}_{t \sim [T]} \mathbb{E} \left\| \mathbb{E}_{\mathbf{y}_t} \nabla F(\mathbf{y}_t) \right\|_1 \leq \frac{\Delta}{DT} + \frac{2 \|\mathbf{G} + \boldsymbol{\sigma}\|_1}{(1 - \beta)T} + \|\boldsymbol{\sigma}\|_1 \sqrt{1 - \beta}$$

$$+ \frac{1}{DT} \left[ \beta \cdot \mathbb{E} \left[ \text{Regret}_T^{[\beta]}(\mathbf{u}_T) \right] + (1 - \beta) \cdot \sum_{t=1}^{T} \mathbb{E} \left[ \text{Regret}_t^{[\beta]}(\mathbf{u}_t) \right] \right] ,$$

*where $\mathbf{y}_t$ is distributed over $\{\mathbf{x}_s\}_{s=1}^{t}$ as $\mathbb{P}(\mathbf{y}_t = \mathbf{x}_s) = \beta^{t-s} \cdot \frac{1-\beta}{1-\beta^t}$ for $s = 1, 2, \ldots, t$.*

Next, let us consider the (expected) regret bound. Fix a coordinate $i = 1, \ldots, d$. Then, by the one-dimensional version of Theorem 9 together with Jensen's inequality, we have the following regret bound for any $t = 1, 2, \ldots, T$:

$$\mathbb{E}\left[\text{Regret}_t^{[\beta]}(\mathbf{u}_t[i])\right] \leq 4D \sqrt{\sum_{t=1}^{T} \beta^{2(T-t)} \mathbb{E}\left|\mathbf{g}_t[i]\right|^2} \leq \frac{4D(G_i + \sigma_i)}{\sqrt{1-\beta}} .$$

Hence, taking the sum over all coordinates $i = 1, \ldots, d$, we obtain

$$\mathbb{E}\left[\text{Regret}_t^{[\beta]}(\mathbf{u}_t)\right] \leq \frac{4D \left\|\boldsymbol{G} + \boldsymbol{\sigma}\right\|_1}{\sqrt{1-\beta}} . \tag{2}$$

Combining these together, one get the following guarantee in terms of the $L_1$ norm. See Section B.2 for a proof.

---

**Theorem 18.** *Suppose that $F$ satisfies Assumption 14 and consider any $\lambda > 0$. For $C > 0$, choose the coordinate-wise optimizer* CLIPPED-ADAM *in Algorithm 1 with the following parameters:*

$$\beta = 1 - \left(\frac{\varepsilon}{10C}\right)^2, \quad D = \frac{(1-\beta)\varepsilon^{1/2}}{4d^{1/2}\lambda^{1/2}}, \quad \text{and} \quad T = \frac{1}{1-\beta} \cdot \max\left\{\frac{4\Delta d^{1/2}\lambda^{1/2}}{\varepsilon^{3/2}}, \frac{12C}{\varepsilon}\right\} .$$

*Then we have $\mathbb{E}_{t \sim [T]} \mathbb{E} \left\|\nabla F(\overline{\mathbf{x}}_t)\right\|_1^{[\lambda]} \leq (1 + \frac{\left\|\boldsymbol{G}+\boldsymbol{\sigma}\right\|_1}{C})\varepsilon$. In other words, a randomly chosen **model EMA** $\overline{\mathbf{x}}_t$ is a $(\lambda, (1 + \frac{\left\|\boldsymbol{G}+\boldsymbol{\sigma}\right\|_1}{C})\varepsilon)$-$L_1$-stationary point, in expectation.*

---

## 6.3 Benefits of coordinate-wise adaptivity of CLIPPED-ADAM

In this section, we discuss the benefits of coordinate-wise adaptivity by examining the guarantee from Theorem 18 and compare it with that of Theorem 11. We begin with the $(\lambda, \varepsilon)$-$L_1$-stationary point guarantee due to Theorem 18. We consider the scenario where $\beta$ is carefully tuned by making the optimal choice of $C$.

**Corollary 19.** *In Theorem 18, choosing $C = \left\|\boldsymbol{G} + \boldsymbol{\sigma}\right\|_1$ leads to the following iteration complexity for finding a $(\lambda, \varepsilon)$-$L_1$-stationary point:*

$$O\left(\max\left\{\frac{\left\|\boldsymbol{G}+\boldsymbol{\sigma}\right\|_1^2 \Delta d^{1/2}\lambda^{1/2}}{\varepsilon^{7/2}}, \frac{\left\|\boldsymbol{G}+\boldsymbol{\sigma}\right\|_1^3}{\varepsilon^3}\right\}\right) .$$

In order to better appreciate the benefits of coordinate-wise adaptivity, let us compare the above iteration complexity with that of Theorem 11.

For concreteness, we treat $G = \left\|\boldsymbol{G}\right\|_2$ and $\sigma = \left\|\boldsymbol{\sigma}\right\|_2$ as constants throughout, and more importantly, we assume that the **coordinates are heterogeneous** in the sense that

$$\left\|\boldsymbol{G}+\boldsymbol{\sigma}\right\|_1 \approx \left\|\boldsymbol{G}+\boldsymbol{\sigma}\right\|_2 . \tag{3}$$

The assumption (3) roughly says that a few coordinates of $\boldsymbol{G} + \boldsymbol{\sigma}$ take much larger values than the rest; if all the coordinates of $\boldsymbol{G} + \boldsymbol{\sigma}$ have similar magnitudes, then $\left\|\boldsymbol{G}+\boldsymbol{\sigma}\right\|_1 \approx \sqrt{d} \left\|\boldsymbol{G}+\boldsymbol{\sigma}\right\|_2$. In the case $\lambda \gtrsim \varepsilon$, Corollary 19 implies that the iteration complexity is

$$O(\left\|\boldsymbol{G}+\boldsymbol{\sigma}\right\|_1^2 \Delta d^{1/2}\lambda^{1/2}\varepsilon^{-7/2}) . \tag{4}$$

Next, let us consider the counterpart that does not adapt to each coordinate separately. In this case, we apply Lemma 16, which tells us that to find a $(\lambda, \varepsilon)$-$L_1$ stationary point it suffices to find a $(\lambda/\sqrt{d}, \varepsilon/\sqrt{d})$-stationary point. Then, from Corollary 12, the iteration complexity is $O(\left\|\boldsymbol{G}+\boldsymbol{\sigma}\right\|_2^2 \Delta(\lambda/\sqrt{d})^{1/2}(\varepsilon/\sqrt{d})^{-7/2})$, i.e.,

$$O(\left\|\boldsymbol{G}+\boldsymbol{\sigma}\right\|_2^2 \Delta d^{3/2}\lambda^{1/2}\varepsilon^{-7/2}) . \tag{5}$$

Hence, when (3) holds, (4) can be lower than (5) by a multiplicative factor of $d$, showing the benefits of coordinate-wise adaptivity.

# 7 Discussion

Our analyses of Adam based on the discounted-to-online conversion is quite different than the previous ones. As discussed in Section 1.1, the previous analyses often result in guarantees that are not quite reflective of practice—*e.g.*, the rates get better without momentum and the rates are no better than that of non-adaptive methods. In contrast, our analyses and results highlight the role of the practical components as highlighted below.

- **Momentum.** In order to obtain a low discounted regret, any sensible online learner should integrate the past history of stochastic gradients $\mathbf{g}_{1:t}$ to make the decision $\mathbf{z}_{t+1}$. Such online learners under the discounted-to-online conversion lead to momentum methods that integrate $\mathbf{g}_{1:t}$ to obtain the next increment $\mathbf{z}_{t+1}$. In particular, non-momentum methods would correspond to aggressive online learners that only use the last gradient $\mathbf{g}_t$ to make the decision $\mathbf{z}_{t+1}$. This perspective provides new insights into understanding the role of momentum, as echoed by Ahn et al. [2024b].

- **Adaptive learning rates.** The adaptive learning rate due to $\beta$-FTRL leads to a gradient-adaptive regret bound (Theorem 9), which is important to obtain a better Lipshitzness dependence (Section 5.2) as well as the coordinate-wise adaptivity for high-dimension settings (Section 6.3). Our analysis offers theoretical benefits of adaptive learning rate from a discounted regret perspective.

- **Model EMA.** Lastly, the discounted-to-nonconvex conversion (Algorithm 1) naturally leads to guarantees in terms of the model EMA, $\overline{\mathbf{x}}_t$. At a high level (see Appendix A precise details), this is because for a dynamic environment, it is important to discount the losses such that online learners adapt to changing environments. The appearance of model EMA in the discounted-to-nonconvex conversion provides a new perspective on its role.

Our analyses and results have several limitations and raise several interesting questions. Firstly, CLIPPED-ADAM does not precisely match the original Adam algorithm, warranting further investigation into the original Adam update. Specifically, our analysis suggests choosing $\beta_1 = \beta$ and $\beta_2 = \beta^2$, which does not align with the commonly used practical choices. Understanding the exact roles of these practical choices for $\beta_1$ and $\beta_2$ would be valuable.

In Section 5.2, we observed that our iteration complexity for finding a $(\lambda, \varepsilon)$-stationary point is $O(\Delta(G + \sigma)^{7/2}\lambda^{1/2}\varepsilon^{-7/2})$ when $G$ and $\sigma$ are unknown. Investigating whether this complexity is optimal presents another intriguing direction for future research.

Lastly, from a practical standpoint, developing a more advanced online learner for discounted regret and designing an algorithm that surpasses Adam in practicality would have significant practical implications.

**Funding Acknowledgments**

AC is supported by NSF grant number CCF-2211718.

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

# Appendix

## A   Proof of discounted-to-nonconvex conversion (Lemma 7)

Note first that via a change of summation, we get

$$
\sum_{n=1}^{T}\sum_{t=1}^{n}\beta^{n-t}(1-\beta)(F(\mathbf{x}_t)-F(\mathbf{x}_{t-1})) = \sum_{t=1}^{T}\sum_{n=t}^{T}\beta^{n-t}(1-\beta)(F(\mathbf{x}_t)-F(\mathbf{x}_{t-1}))
$$

$$
= \sum_{t=1}^{T}(1-\beta^{T-t+1})(F(\mathbf{x}_t)-F(\mathbf{x}_{t-1}))
$$

$$
= F(\mathbf{x}_T)-F(\mathbf{x}_0) - \sum_{t=1}^{T}\beta^{T-t+1}(F(\mathbf{x}_t)-F(\mathbf{x}_{t-1})).
$$

Rearranging the above together with the fact $F(\mathbf{x}_0) - F(\mathbf{x}_T) \le F(\mathbf{x}_0) - \inf_{\mathbf{x}} F(\mathbf{x}) =: \Delta$, we get

$$
-\Delta \le \underbrace{\mathbb{E}\left[\sum_{n=1}^{T}\sum_{t=1}^{n}\beta^{n-t}(1-\beta)(F(\mathbf{x}_t)-F(\mathbf{x}_{t-1}))\right]}_{\text{A}} + \underbrace{\mathbb{E}\left[\sum_{t=1}^{T}\beta^{T-t+1}(F(\mathbf{x}_t)-F(\mathbf{x}_{t-1}))\right]}_{\text{B}}.
$$

We also recall the following fact about exponential random variable due to [Zhang and Cutkosky, 2024, Lemma 3.1].

**Lemma 20.** *Let $\alpha \sim Exp(\lambda)$ for some $\lambda > 0$, then*

$$
\mathbb{E}_{\alpha}[F(\mathbf{x}+\alpha\mathbf{z}) - F(\mathbf{x})] = \mathbb{E}_{\alpha}[\langle \nabla F(\mathbf{x}+\alpha\mathbf{z}), \mathbf{z}\rangle]/\lambda.
$$

By Lemma 20 with $\lambda = 1$, together with $\mathbf{x}_t = \mathbf{x}_{t-1} + \alpha_t \mathbf{z}_t$, it follows that

$$
\mathbb{E}[F(\mathbf{x}_t) - F(\mathbf{x}_{t-1})] = \mathbb{E}\left[\langle \mathbf{g}_t, \mathbf{z}_t\rangle\right].
$$

This identity indicates that the function gap is exactly equal to the linearization of the function gap. In this sense, the randomization renders the first-order Taylor approximation perfectly accurate.

Now, with this result, we will address each term separately.

### Analysis of Ⓐ

Note that for each $t \le n$, since $\mathbf{x}_t = \mathbf{x}_{t-1} + \alpha_t \mathbf{z}_t$ for $\alpha_t \sim \text{Exp}(1)$, Lemma 20 yields

$$
\mathbb{E}[F(\mathbf{x}_t) - F(\mathbf{x}_{t-1})] = \mathbb{E}\langle \nabla F(\mathbf{x}_t), \mathbf{z}_t\rangle = \mathbb{E}\langle \nabla F(\mathbf{x}_t), \mathbf{u}_n\rangle + \mathbb{E}\langle \nabla F(\mathbf{x}_t), \mathbf{z}_t - \mathbf{u}_n\rangle
$$

$$
= \mathbb{E}\langle \nabla F(\mathbf{x}_t), \mathbf{u}_n\rangle + \mathbb{E}\langle \nabla F(\mathbf{x}_t) - \mathbf{g}_t, \mathbf{z}_t - \mathbf{u}_n\rangle + \mathbb{E}\langle \mathbf{g}_t, \mathbf{z}_t - \mathbf{u}_n\rangle
$$

$$
= \underbrace{\mathbb{E}\langle \nabla F(\mathbf{x}_t), \mathbf{u}_n\rangle}_{①} + \underbrace{\mathbb{E}\langle \nabla F(\mathbf{x}_t) - \mathbf{g}_t, -\mathbf{u}_n\rangle}_{②} + \underbrace{\mathbb{E}\langle \mathbf{g}_t, \mathbf{z}_t - \mathbf{u}_n\rangle}_{③},
$$

where the last line follows from the fact $\mathbb{E}[\langle \nabla F(\mathbf{x}_t) - \mathbf{g}_t, \mathbf{z}_t \rangle] = 0$. More specifically, note that the randomness in the stochastic gradient oracle is independent of the randomness due to $\alpha_t$. Since $\mathbb{E}[\nabla F(\mathbf{x}_t) - \mathbf{g}_t] = 0$, it follows that

$$\mathbb{E}[\langle \nabla F(\mathbf{x}_t) - \mathbf{g_t}, \mathbf{z_t} \rangle] = \mathbb{E}[\langle \mathbb{E}[\nabla F(\mathbf{x}_t) - \mathbf{g_t}], \mathbf{z_t} \rangle] = 0,$$

where the inner expectation is with respect to the randomness in the stochastic gradient oracle and the outer is with respect to all other quantities.

Now let us handle each term.

$\boxed{1}$: Note that using the definition of $\mathbf{y}_n$, we have

$$\mathbb{E}\sum_{t=1}^{n}\beta^{n-t}(1-\beta)\left\langle \nabla F(\mathbf{x}_t), \mathbf{u}_n\right\rangle = (1-\beta)\,\mathbb{E}\left\langle \sum_{t=1}^{n}\beta^{n-t}\nabla F(\mathbf{x}_t), -D\frac{\sum_{t=1}^{n}\beta^{n-t}\nabla F(\mathbf{x}_t)}{\|\sum_{t=1}^{n}\beta^{n-t}\nabla F(\mathbf{x}_t)\|}\right\rangle$$

$$= (1-\beta^n)\,\mathbb{E}\left\langle \sum_{t=1}^{n}\frac{1-\beta}{1-\beta^n}\beta^{n-t}\nabla F(\mathbf{x}_t), -D\frac{\sum_{t=1}^{n}\frac{1-\beta}{1-\beta^n}\beta^{n-t}\nabla F(\mathbf{x}_t)}{\left\|\sum_{t=1}^{n}\frac{1-\beta}{1-\beta^n}\beta^{n-t}\nabla F(\mathbf{x}_t)\right\|}\right\rangle$$

$$= -D(1-\beta^n)\,\mathbb{E}\left\|\mathbb{E}_{\mathbf{y}_n}\nabla F(\mathbf{y}_n)\right\| \le -D\,\mathbb{E}\left\|\mathbb{E}_{\mathbf{y}_n}\nabla F(\mathbf{y}_n)\right\| + DG\beta^n\,.$$

Therefore, summing over $n = 1, \ldots, T$, we obtain:

$$\mathbb{E}\sum_{n=1}^{T}\sum_{t=1}^{n}\beta^{n-t}(1-\beta)\left\langle \nabla F(\mathbf{x}_t), \mathbf{u}_n\right\rangle \le -D\,\mathbb{E}\sum_{t=1}^{T}\left\|\mathbb{E}_{\mathbf{y}_t}\nabla F(\mathbf{y}_t)\right\| + \frac{DG}{1-\beta}\,.$$

$\boxed{2}$: For the second term, using Cauchy-Schwartz inequality, we have

$$\mathbb{E}\sum_{t=1}^{n}\beta^{n-t}\left\langle \nabla F(\mathbf{x}_t) - \mathbf{g}_t, -\mathbf{u}_n\right\rangle \le \sqrt{\mathbb{E}\left\|\sum_{t=1}^{n}\beta^{n-t}(\nabla F(\mathbf{x}_t) - \mathbf{g}_t)\right\|^2 \mathbb{E}\left\|\mathbf{u}_n\right\|^2}\,.$$

Using the bounded variance assumption on the stochastic gradient oracle, we have

$$\mathbb{E}\left\|\sum_{t=1}^{n}\beta^{n-t}(\nabla F(\mathbf{x}_t) - \mathbf{g}_t)\right\|^2 = \mathbb{E}\sum_{t=1}^{n}\beta^{2(n-t)}\left\|\nabla F(\mathbf{x}_t) - \mathbf{g}_t\right\|^2 \le \frac{\sigma^2}{1-\beta^2}\,.$$

Therefore, summing over $n = 1, \ldots, T$, and using the fact that $\frac{1}{1-\beta^2} \le \frac{1}{1-\beta}$, we get the following bound on the second term:

$$\mathbb{E}\sum_{n=1}^{T}\sum_{t=1}^{n}\beta^{n-t}(1-\beta)\left\langle \nabla F(\mathbf{x}_t) - \mathbf{g}_t, -\mathbf{u}_n\right\rangle \le \sum_{n=1}^{T}(1-\beta)\cdot\frac{\sigma D}{\sqrt{1-\beta^2}} \le \sigma DT\sqrt{1-\beta}\,.$$

$\boxed{3}$: Lastly, for the third term, we have

$$\mathbb{E}\sum_{n=1}^{T}\sum_{t=1}^{n}\beta^{n-t}(1-\beta)\left\langle \mathbf{g}_t, \mathbf{z}_t - \mathbf{u}_n\right\rangle = (1-\beta)\,\mathbb{E}\sum_{n=1}^{T}\left[\sum_{t=1}^{n}\mathbb{E}\left\langle \beta^{n-t}\mathbf{g}_t, \mathbf{z}_t - \mathbf{u}_n\right\rangle\right]$$

$$= (1-\beta)\,\mathbb{E}\sum_{t=1}^{T}\mathrm{Regret}_t^{[\beta]}(\mathbf{u}_t)\,.$$

**Analysis of $\boxed{\text{B}}$**

Note that for each $t$, since $\mathbf{x}_t \leftarrow \mathbf{x}_{t-1} + \alpha_t\mathbf{z}_t$ for $\alpha_t \sim \mathrm{Exp}(1)$, Lemma 20 yields

$$\mathbb{E}[F(\mathbf{x}_t) - F(\mathbf{x}_{t-1})] = \mathbb{E}\left\langle \nabla F(\mathbf{x}_t), \mathbf{z}_t\right\rangle = \mathbb{E}\left\langle \mathbf{g}_t, \mathbf{z}_t\right\rangle$$

$$= \mathbb{E}\left\langle \mathbf{g}_t, \mathbf{z}_t - \mathbf{u}_T\right\rangle + \mathbb{E}\left\langle \mathbf{g}_t, \mathbf{u}_T\right\rangle \le \mathbb{E}\left\langle \mathbf{g}_t, \mathbf{z}_t - \mathbf{u}_T\right\rangle + D(G + \sigma)\,.$$

Thus,

$$\mathbb{E}\left[\sum_{t=1}^{T}\beta^{T-t+1}(F(\mathbf{x}_t)-F(\mathbf{x}_{t-1}))\right] = \beta\,\mathbb{E}\sum_{t=1}^{T}\left[\langle\beta^{T-t}\mathbf{g}_t,\mathbf{z}_t-\mathbf{u}_T\rangle+\beta^{T-t}D(G+\sigma)\right]$$

$$\leq \beta\,\mathbb{E}[\mathrm{Regret}_T^{[\beta]}(\mathbf{u}_T)]+\frac{D(G+\sigma)}{1-\beta}\,.$$

## Combining Ⓐ and Ⓑ

Combining the above analyses and rearranging, it follows that

$$D\,\mathbb{E}\sum_{t=1}^{T}\left\|\underset{\mathbf{y}_t}{\mathbb{E}}\,\nabla F(\mathbf{y}_t)\right\| \leq \Delta+\frac{DG}{1-\beta}+\sigma DT\sqrt{1-\beta}+(1-\beta)\,\mathbb{E}\sum_{t=1}^{T}\left[\mathrm{Regret}_t^{[\beta]}(\mathbf{u}_t)\right]$$

$$+\beta\,\mathbb{E}[\mathrm{Regret}_T^{[\beta]}(\mathbf{u}_T)]+\frac{D(G+\sigma)}{1-\beta}\,.$$

Dividing both sides by $DT$, we get the desired result.

### A.1 Proof of the coordinate-wise version (Lemma 17)

The proof closely follows that of Lemma 7. In particular, with $\Delta := F(\mathbf{x}_0)-\inf_{\mathbf{x}}F(\mathbf{x})$, we have

$$-\Delta \leq \underbrace{\mathbb{E}\left[\sum_{n=1}^{T}\sum_{t=1}^{n}\beta^{n-t}(1-\beta)(F(\mathbf{x}_t)-F(\mathbf{x}_{t-1}))\right]}_{\text{Ⓐ}}+\underbrace{\mathbb{E}\left[\sum_{t=1}^{T}\beta^{T-t+1}(F(\mathbf{x}_t)-F(\mathbf{x}_{t-1}))\right]}_{\text{Ⓑ}}.$$

We begin with the term Ⓑ. Using the same decomposition as before, we have

$$\mathbb{E}[F(\mathbf{x}_t)-F(\mathbf{x}_{t-1})] = \mathbb{E}\langle\mathbf{g}_t,\mathbf{z}_t-\mathbf{u}_T\rangle+\mathbb{E}\langle\mathbf{g}_t,\mathbf{u}_T\rangle = \mathbb{E}\langle\mathbf{g}_t,\mathbf{z}_t-\mathbf{u}_T\rangle+\sum_{i=1}^{d}\mathbb{E}\,\mathbf{g}_t[i]\mathbf{u}_T[i]$$

$$\leq \mathbb{E}\langle\mathbf{g}_t,\mathbf{z}_t-\mathbf{u}_T\rangle+D\sum_{i=1}^{d}(G_i+\sigma_i)\,.$$

Thus,

$$\mathbb{E}\left[\sum_{t=1}^{T}\beta^{T-t+1}(F(\mathbf{x}_t)-F(\mathbf{x}_{t-1}))\right] = \beta\,\mathbb{E}\sum_{t=1}^{T}\left[\langle\beta^{T-t}\mathbf{g}_t,\mathbf{z}_t-\mathbf{u}_T\rangle+\beta^{T-t}\sum_{i=1}^{d}D(G_i+\sigma_i)\right]$$

$$\leq \beta\,\mathbb{E}[\mathrm{Regret}_T^{[\beta]}(\mathbf{u}_T)]+\frac{D\sum_{i=1}^{d}(G_i+\sigma_i)}{1-\beta}\,.$$

Moving onto the term Ⓐ, we again use the same decomposition:

$$\mathbb{E}[F(\mathbf{x}_t)-F(\mathbf{x}_{t-1})] = \underbrace{\mathbb{E}\langle\nabla F(\mathbf{x}_t),\mathbf{u}_n\rangle}_{\text{①}}+\underbrace{\mathbb{E}\langle\nabla F(\mathbf{x}_t)-\mathbf{g}_t,-\mathbf{u}_n\rangle}_{\text{②}}+\underbrace{\mathbb{E}\langle\mathbf{g}_t,\mathbf{z}_t-\mathbf{u}_n\rangle}_{\text{③}}\,.$$

As before, let us handle each term one by one separately.

(1): Note that using the definition of $\mathbf{y}_n$, for each coordinate $i = 1, \ldots, d$, we have

$$\mathbb{E} \sum_{t=1}^{n} \beta^{n-t}(1-\beta)\partial_i F(\mathbf{x}_t)\mathbf{u}_n[i] = (1-\beta)\,\mathbb{E}\left[\left(\sum_{t=1}^{n} \beta^{n-t}\nabla F(\mathbf{x}_t)\right)\left(-D\frac{\sum_{t=1}^{n} \beta^{n-t}\partial_i F(\mathbf{x}_t)}{|\sum_{t=1}^{n} \beta^{s-t}\partial_i F(\mathbf{x}_t)|}\right)\right]$$

$$= (1-\beta^n)\,\mathbb{E}\left[\left(\sum_{t=1}^{n} \frac{1-\beta}{1-\beta^n}\beta^{n-t}\nabla F(\mathbf{x}_t)\right)\left(-D\frac{\sum_{t=1}^{n} \frac{1-\beta}{1-\beta^n}\beta^{n-t}\nabla F(\mathbf{x}_t)}{\left|\sum_{t=1}^{n} \frac{1-\beta}{1-\beta^n}\beta^{n-t}\nabla F(\mathbf{x}_t)\right|}\right)\right]$$

$$= -D(1-\beta^n)\,\mathbb{E}\left|\mathop{\mathbb{E}}_{\mathbf{y}_n} \partial_i F(\mathbf{y}_n)\right| \le -D\,\mathbb{E}\left|\mathop{\mathbb{E}}_{\mathbf{y}_n} \partial_i F(\mathbf{y}_n)\right| + DG_i\beta^n.$$

Therefore, summing over $i = 1, \ldots, d$ and then $n = 1, \ldots, T$, we obtain:

$$\mathbb{E} \sum_{n=1}^{T}\sum_{t=1}^{n} \beta^{n-t}(1-\beta)\langle \nabla F(\mathbf{x}_t), \mathbf{u}_n\rangle \le -D\,\mathbb{E}\sum_{t=1}^{T}\left\|\mathop{\mathbb{E}}_{\mathbf{y}_t}\nabla F(\mathbf{y}_t)\right\|_1 + \frac{D\sum_{i=1}^{d} G_i}{1-\beta}.$$

(2): For each coordinate $i = 1, \ldots, d$, we have

$$\mathbb{E} \sum_{t=1}^{n} \beta^{n-t}(\partial_i F(\mathbf{x}_t) - \mathbf{g}_t[i])(-\mathbf{u}_n[i]) \le \sqrt{\mathbb{E}\left|\sum_{t=1}^{n} \beta^{n-t}(\partial_i F(\mathbf{x}_t) - \mathbf{g}_t[i])\right|^2 \mathbb{E}\,|\mathbf{u}_n[i]|^2}.$$

Using the coordinate-wise bounded variance assumption on the stochastic gradient oracle,

$$\mathbb{E}\left|\sum_{t=1}^{n} \beta^{n-t}(\partial_i F(\mathbf{x}_t) - \mathbf{g}_t[i])\right|^2 = \mathbb{E}\sum_{t=1}^{n} \beta^{2(n-t)}|\partial_i F(\mathbf{x}_t) - \mathbf{g}_t[i]|^2 \le \frac{\sigma_i^2}{1-\beta^2}.$$

Therefore, summing over $n = 1, \ldots, T$, and using the fact that $\frac{1}{1-\beta^2} \le \frac{1}{1-\beta}$, we get the following bound on the second term:

$$\mathbb{E} \sum_{n=1}^{T}\sum_{t=1}^{n} \beta^{n-t}(1-\beta)\langle \nabla F(\mathbf{x}_t) - \mathbf{g}_t, -\mathbf{u}_n\rangle \le \sum_{n=1}^{T}(1-\beta)\cdot\frac{D\sum_{i=1}^{d}\sigma_i}{\sqrt{1-\beta^2}}$$

$$\le DT\left(\sum_{i=1}^{d}\sigma_i\right)\sqrt{1-\beta}.$$

(3): We use the same manipulation as before:

$$\mathbb{E}\sum_{n=1}^{T}\sum_{t=1}^{n}\beta^{n-t}(1-\beta)\langle\mathbf{g}_t, \mathbf{z}_t - \mathbf{u}_n\rangle = (1-\beta)\,\mathbb{E}\sum_{t=1}^{T}\mathrm{Regret}_t^{[\beta]}(\mathbf{u}_t).$$

Combining the above, we get the desired result in Lemma 17.

## B    Proof of main theorems

### B.1    Proof of Theorem 11

By Definition 3, since $\mathbb{E}[\mathbf{y}_t] = \bar{\mathbf{x}}_t$, it holds that

$$\mathop{\mathbb{E}}_{t\sim[T]}\|\nabla F(\bar{\mathbf{x}}_t)\|^{[\lambda]} \le \mathop{\mathbb{E}}_{t\sim[T]}\left[\left\|\mathop{\mathbb{E}}_{\mathbf{y}_t}\nabla F(\mathbf{y}_t)\right\| + \lambda\mathop{\mathbb{E}}_{\mathbf{y}_t}\|\mathbf{y}_t - \bar{\mathbf{x}}_t\|^2\right].$$

We begin with the second term (the variance term). By Lemma 10, we have

$$\lambda\mathop{\mathbb{E}}_{t\sim[T]}\mathop{\mathbb{E}}_{\mathbf{y}_t}\|\mathbf{y}_t - \bar{\mathbf{x}}_t\|^2 \le 12\frac{\lambda D^2}{(1-\beta)^2}.$$

Hence, by choosing $D = \frac{(1-\beta)\varepsilon^{1/2}}{4\lambda^{1/2}}$, it follows that $\lambda \cdot \mathbb{E}_{t\sim[T]}\,\mathbb{E}_{\mathbf{y}_t}\left\|\mathbf{y}_t - \overline{\mathbf{x}}_t\right\|^2 \le \varepsilon$.

Next consider the first term (the norm of the averaged gradients). Plugging the regret bound (1) into Lemma 7, we get

$$
\mathop{\mathbb{E}}_{t\sim[T]}\left\|\mathop{\mathbb{E}}_{\mathbf{y}_t}\nabla F(\mathbf{y}_t)\right\| \le \frac{\Delta}{DT} + \frac{2G+\sigma}{(1-\beta)T} + \sigma\sqrt{1-\beta} + \frac{4(G+\sigma)}{T\sqrt{1-\beta}} + 4(G+\sigma)\sqrt{1-\beta}
$$

$$
\le \frac{4\Delta\lambda^{1/2}}{(1-\beta)\varepsilon^{1/2}T} + \frac{6G+5\sigma}{(1-\beta)T} + (4G+5\sigma)\sqrt{1-\beta}\,,
$$

where the last line follows since $\frac{1}{\sqrt{1-\beta}} \le \frac{1}{1-\beta}$ and $D = \frac{(1-\beta)\varepsilon^{1/2}}{4\lambda^{1/2}}$. Choosing $\beta = 1 - (\frac{\varepsilon}{10C})^2$, the last term is bounded by $\frac{G+\sigma}{2C}\varepsilon$. Moreover, choosing $T = (1-\beta)^{-1}\cdot\max\left\{4\Delta\lambda^{1/2}\varepsilon^{-3/2},\ 12C\varepsilon^{-1}\right\}$, the first and second terms are bounded by $\varepsilon$ and $\frac{G+\sigma}{2C}\varepsilon$, respectively. This concludes the proof.

## B.2 Proof of Theorem 18

By Definition 15, since $\mathbb{E}[\mathbf{y}_t] = \overline{\mathbf{x}}_t$, it holds that

$$
\mathop{\mathbb{E}}_{t\sim[T]}\|\nabla F(\overline{\mathbf{x}}_t)\|_1^{[\lambda]} \le \mathop{\mathbb{E}}_{t\sim[T]}\left[\left\|\mathop{\mathbb{E}}_{\mathbf{y}_t}\nabla F(\mathbf{y}_t)\right\|_1 + \lambda\mathop{\mathbb{E}}_{\mathbf{y}_t}\|\mathbf{y}_t - \overline{\mathbf{x}}_t\|_2^2\right].
$$

We begin with the second term (the variance term). This time, given that now each coordinate of update $\mathbf{z}_t$ is bounded by $D$, i.e., $|\mathbf{z}_t[i]| \le D$, applying the variance bound due to Lemma 10 coordinate-wise implies:

$$
\lambda\mathop{\mathbb{E}}_{t\sim[T]}\mathop{\mathbb{E}}_{\mathbf{y}_t}\|\mathbf{y}_t - \overline{\mathbf{x}}_t\|_2^2 \le 12\frac{\lambda dD^2}{(1-\beta)^2}\,.
$$

Hence, by choosing $D = \frac{(1-\beta)\varepsilon^{1/2}}{4d^{1/2}\lambda^{1/2}}$, it follows that $\lambda\mathbb{E}_{t\sim[T]}\,\mathbb{E}_{\mathbf{y}_t}\left\|\mathbf{y}_t - \overline{\mathbf{x}}_t\right\|^2 \le \varepsilon$.

Next consider the first term (the $L_1$-norm of the averaged gradients). Plugging the regret bound (2) into Lemma 17, and doing similar manipulations as the proof of Theorem 11, we get

$$
\mathop{\mathbb{E}}_{t\sim[T]}\left\|\mathop{\mathbb{E}}_{\mathbf{y}_t}\nabla F(\mathbf{y}_t)\right\|_1 \le \frac{\Delta}{DT} + \frac{\|6\boldsymbol{G}+5\boldsymbol{\sigma}\|_1}{(1-\beta)T} + \|4\boldsymbol{G}+5\boldsymbol{\sigma}\|_1\sqrt{1-\beta}
$$

$$
= \frac{4\Delta d^{1/2}\lambda^{1/2}}{(1-\beta)\varepsilon^{1/2}T} + \frac{\|6\boldsymbol{G}+5\boldsymbol{\sigma}\|_1}{(1-\beta)T} + \|4\boldsymbol{G}+5\boldsymbol{\sigma}\|_1\sqrt{1-\beta}\,,
$$

where the last line follows since $D = \frac{(1-\beta)\varepsilon^{1/2}}{4d^{1/2}\lambda^{1/2}}$. Choosing $\beta = 1 - (\frac{\varepsilon}{10C})^2$, the last term is bounded by $\frac{\|\boldsymbol{G}+\boldsymbol{\sigma}\|_1}{2C}\cdot\varepsilon$. Moreover, choosing $T = (1-\beta)^{-1}\cdot\max\left\{4\Delta d^{1/2}\lambda^{1/2}\varepsilon^{-3/2},\ 12C\varepsilon^{-1}\right\}$, the first and second terms are bounded by $\varepsilon$ and $\frac{\|\boldsymbol{G}+\boldsymbol{\sigma}\|_1}{2C}\cdot\varepsilon$, respectively. This concludes the proof.

