# OpenReview forum: "Adam with model exponential moving average is effective for nonconvex optimization"
_NeurIPS.cc/2024/Conference — NeurIPS 2024 poster_

### Official Review · Reviewer_m9Ez · 2024-07-08

**Soundness:** 3
**Presentation:** 2
**Contribution:** 2
**Rating:** 4
**Confidence:** 2

**Summary:**

In this paper, the authors analyze the online learning framework 'discounted-to-nonconvex conversion', and propose a bound for the expectation of the exponential average of the output gradient. Moreover, they find that if the online learner follows a specific loss structure(FTRL), then the output of this online learner would follow a clip-Adam rules.

**Strengths:**

The result of this paper is solid.

**Weaknesses:**

Since this paper is built upon the 'discounted-to-nonconvex conversion' framework proposed in [1], focusing on the perspective of online learning, many definitions, concepts, and details require further elaborations and interpretations, especially for readers who are unfamiliar with this topic. I will list all my confusions and questions in the 'Questions' part and update my rating according to the response from the authors.

[1] Qinzi Zhang and Ashok Cutkosky. Random scaling and momentum for non-smooth non-convex optimization. In International Conference on Machine Learning. PMLR, 2024.

**Questions:**

1. In line 109, the description of the Algorithm 1, could the author explain the reason they introduce the notations $\mathbf{x_t}$ and $\mathbf{w_t}$, since from my perspective, $\mathbf{x_t}$ represent the iterations of coefficients of the loss function. Then the stochastic gradient $\mathbf{g_t}$ should also be computed by $\mathbf{x_t}$, instead of $\mathbf{w_t}$. I also wonder about the implication of $\alpha_t$ in the definition of $\mathbf{w_t}$, since it only benefits the further proof.

2. I might be wrong, but there might be typos in the formula of Definition 6. According to the definition of $\ell_t^{[\beta]}(z)$ in Algorithm 1,  the RHS of formula might be $\beta^t\sum_{s=1}^t \langle \beta^{-s} \mathbf{v_s},  \mathbf{z_s} -\mathbf{\mu}\rangle$. Meanwhile, I also wonder whether $\mathbf{v_s}$ is actually \mathbf{g_s} defined in Algorithm 1 since I never find the definition of $\mathbf{v_s}$. Moreover, could the author explain the meanings and implications of such a definition?

3. Could the author explain the meaning of choosing such a comparator $u_t$ in Lemma 7.

4. I might be wrong, but as I checked the Lagrangian function of constrained optimization in line 137, I guess the RHS should be $-\text{clip}_D(\eta \sum \mathbf{v_s})$.

5. Could the authors explain the details of how they get Corollary 12 from the results of the definition $T$ in Theorem 11.

6. Could the authors explain why line 466 holds since I'm not sure about the independence among $\mathbf{z_t}$ and $\mathbf{g_t}$. Moreover, could the author explain about the randomness in their formulas and which variables they take expectation w.r.t during the proof?

7. Could the author further discuss the implications of their main result? Since from my perspective, it looks like they use the exponential discounted loss and choose FTRL learner in Discounted-to-nonconvex conversion, then they could get an updating rules share a similar structure with Adam. However, this result is far-fetched, and I have no idea what it implies.

**Limitations:**

No limitations.

---

> ### Author Rebuttal · Authors · 2024-08-03
>
> Thank you for your detailed comments! We answer your questions one by one below.
>
> 1. You are correct that $\mathbf{g_t}$ is computed at $\mathbf{x_t}$ in practice. **In our new version, we actually fix this issue**, and in our new main results, $\mathbf{g_t}$ is computed at $\mathbf{x_t}$, and there is no need to introduce $\mathbf{w_t}$ anymore. This is based on the "exponential scaling" technique due to [Zhang and Cutkosky, 2024], and we will update this in our final version.
>
>    - The role of $\alpha_t$ is for the identity $F(\mathbf{x_{t}}) - F(\mathbf{x_{t-1}}) \leq \mathbb{E}[\langle \mathbf{g_t},~\mathbf{x_t}-\mathbf{x_{t-1}} \rangle]$ to hold, which is crucial for the discounted-to-nonconvex conversion to work. When $F$ is convex, this holds without the random scaling $\alpha_t$, but it turns out for the nonconvex setting, we need this random scaling. In our final version, where we fix the afornmentioned issue regarding $\mathbf{w}_t$, we choose $\alpha_t$ to be an exponential random variable, inspired by [Zhang and Cutkosky, 2024].
>
>    - Regarding $\alpha$ being beneficial for the further proofs, sorry it's an overload of notations; the $\alpha$'s that appears in the later part of the analysis should've been a separate quantity. We changed $\alpha$'s in the later part to $C$ in our final version.
>
> 2. Thank you for catching the typos. It should be ineed $\beta^t\sum_{s=1}^t \langle \beta^{-s} \mathbf{g_s},  \mathbf{z_s} -\mathbf{u}\rangle$. We corrected it in our final version.
>
> 3. The comparator $\mathbf{u}_t$ roughly models the "update direction an oracle algorithm with perfect knowledge of the loss would have chosen." In the proof of Lemma 7, we show that along such a direction, the loss value decreases, and that is how we could come up with the convergence guarantee.
>
>
> 4. You are correct. Thanks for catching the typo.
>
> 5. Let's start with $$T = \frac{1}{1-\beta} \max\{\frac{5\Delta \lambda^{1/2}}{\epsilon^{3/2}}, \frac{12\alpha}{\epsilon}\}$$ in Theorem 11.
> SInce $1-\beta = (\frac{\epsilon}{10\alpha})^2$ and $\alpha=G+\sigma$, it follows that
>  $$T = \frac{100\alpha^2}{\epsilon^2} \max\{\frac{5\Delta \lambda^{1/2}}{\epsilon^{3/2}}, \frac{12\alpha}{\epsilon}\} = O\left(   \max\{\frac{(G+\sigma)^2\Delta \lambda^{1/2}}{\epsilon^{7/2}}, \frac{(G+\sigma)^3}{\epsilon^3}\} \right)$$
>  as desired.
>
> 6. Thank you for your question. We will detail the argument in our final version. In our theorem statements, the expectations are taken over both randomness in the gradient oracle as well as the internal randomness of the algorithm from choosing $\alpha$. Note that the randomness in the stochastic gradient oracle is independent of the randomness due to $\alpha_t$. Since $\mathbb{E}[\nabla F(\mathbf{w_t})- \mathbf{g_t}] =0$, it follows that
> $$\mathbb{E}[\langle \nabla F(\mathbf{w_t})- \mathbf{g_t}, \mathbf{z_t} \rangle]   = \mathbb{E}[ \langle \mathbb{E}[ \nabla F(\mathbf{w_t})- \mathbf{g_t}], \mathbf{z_t} \rangle] = 0,$$
> where the inner expectation is with respect to the randomness in the stochastic gradient oracle and the outer is with respect to all other quantities.
> In general, we will detail all the arguments regarding expectations in our final version.
>
> 7. As discussed in the introduction section, Adam with EMA is very succesful for large complex neural network models. Our main goal was to understand why such a method is very effective.
> Although we do have a minor modification to the original Adam (such as clipping), our analysis shows that Adam with EMA achieves an optimal convergence rate, and it is particularly benefical when the Lipshitzness constants are unknown and heterogenous across different coordinates.
> We believe our results to be a significant progress towards understanding the success of Adam (with EMA) since before our work, it is unclear from the theoretical standpoint as to why Adam is much more effective than SGD.
>
> We would be glad to address any remaining concern!

---

> ### Comment · Reviewer_m9Ez · 2024-08-13
>
> I thank the authors for their response.
> I still have some resolved questions. Since the author claimed they had dropped the notation of $w_t$, which plays a significant role in their original proof, I can not check the correctness of their results in their new version. Despite this significant problem, I also feel confused about their explanations for the following points:
> - Since they drop the variable $w_t$, why they still need $\alpha_t$ for their further proof? At least in their original version,  $\alpha_t$ was introduced in the definition of $w_t$, and now, I even don't know where is this $\alpha_t$ first introduced.
> - The authors might misunderstand my concerns about line 466. Actually, I want to know why $E[\nabla F(w_t)-g_t|z_t]=0$.
> - The authors's explanation of implications and motivations does not convince me, several definitions or settings, seems too far-fetched and specific. While the authors consider clipping a minor modification, I have never used it or seen it used by others in practice. Similar questions also exist for the definition of comparator $u_t$ and learning rate $\eta$. As I stated in my review, it seems only given such specific forms, the authors get an updating rules share a similar structure with Adam. However, I can not see any direct connections between these frameworks and Adam.
>
> Due to these questions, I have to keep my original rating score. However, I want to stress that I'm not familiar with online optimization. Specifically, my comments about these intermediate variables might be unfair and it's welcome to point it out with more convincing evidence. Moreover, I keep my confidence at 2 and if other reviewers feel more confident in their judgment, I would defer to them.

---

> > ### Author Response · Authors · 2024-08-13
> > **Thank you**
> >
> > - To clarify the new scheme, the update is now $\mathbf{x_t}=  \mathbf{x_{t-1}} + \alpha_t \mathbf{z_t}$, where $\alpha_t$ is sampled from an exponential distribution (which is a very light-tailed distribution with mean 1 and variance 1). So $\alpha_t$ is still used.
> >
> >   Via an elementary argument involving  fundamental theorem of calculus  and the pdf $p(\alpha) = \exp(-\alpha)$,  we obtain $\mathbb{E}[F(\mathbf{x_t}) - F(\mathbf{x_{t-1}})] = \mathbb{E}[\langle \mathbf{g_t}, \mathbf{x_t} -  \mathbf{x_{t-1}}\rangle]$.
> >
> >   This identity says that the function gap is *exactly equal* to the linearization of the function gap. In a sense, the randomization makes the first-order Taylor approximation perfectly correct. In the current submission, we sample $\mathbf{g_t}$ at $\mathbf{w_t}$ to ensure this identity. However, changing $\alpha_t$ to be exponentially distributed and multiplying the update  $\mathbf{z_t}$ by $\alpha_t$ means that  this is no longer necessary.
> >
> >
> >   In short, although $\mathbf{w_t}$ plays a significant role, the entirety of that role is contained in establishing the identity $\mathbb{E}[F(\mathbf{x_t}) - F(\mathbf{x_{t-1}})] = \mathbb{E}[\langle \mathbf{g_t}, \mathbf{x_t} -  \mathbf{x_{t-1}}\rangle]$. This other option keeps the same identity and so works just as well.
> >
> >   That said, if the reviewers prefer to keep the results  the same as in the submitted version for which all of the analysis is completely available, we are of course happy to do so.
> >
> > - Regarding why $\mathbb{E}[ F(\mathbf{w_t}) - \mathbf{g_t}|\mathbf{z_t}]=0$: this is is because $\mathbf{g_t}$ is a standard stochastic gradient oracle evaluated at $\mathbf{w_t}$ and as such $\mathbb{E}[\mathbf{g_t} | \mathbf{w_t}]=F(\mathbf{w_t})$. As a concrete illustration, suppose that $F=\mathbb{E}[f(x,r)]$, where $x$ is the model parameter, $r$ is a randomly sampled data point, and $f$ is the loss of the model on the data point. Then  $\mathbf{g_t}$ can be taken to be $\nabla  f(\mathbf{w_t}, r_t)$  for some i.i.d. sample $r$.  This formalism exactly captures the standard approach used in training. In this case, since $r_t$ is independent of $\mathbf{w_t}$ and $\mathbf{z_t}$, we have $\mathbb{E}[ F(\mathbf{w_t}) - \mathbf{g_t} | \mathbf{z_t}]=0$.
> >
> > - Regarding the clipping: You're right that this is not too common in current practice. However, we feel this is more an analytical artifact than a real change in the algorithm because the clipping should not be active in most iterations. The reason for this is that intuitively as the algorithm convergence we expect the gradients to be dominated by noise so that they look roughly like mean-zero random variables. In this case, the clipping will not occur. On the other hand, if the clipping *does* occur frequently, then an alternative analysis based upon https://jmlr.org/papers/v18/17-079.html would actually show that we converge incredibly fast. This in turn would put us in the regime in which gradients are dominated by noise, and so clipping would stop occurring.

---

### Official Review · Reviewer_DXTx · 2024-07-12

**Soundness:** 3
**Presentation:** 2
**Contribution:** 2
**Rating:** 5
**Confidence:** 3

**Summary:**

The authors show that with clipping and model exponential average, (Clipping) Adam can perform better than SGD.

**Strengths:**

Give the analysis of (Clipping) Adam with model exponential average.

**Weaknesses:**

1.  The paper is hard to follow.

    (i) In Algorithm 1, it would be better to add "output: $\bar{w}_T$"

    (ii) it is unclear what $v_s$ is in Definition 6.

2. All of the following results are based on Lemma 7, I am not sure whether Lemma 7 is suitable for SGD.

**Questions:**

If change Lemma 7 with a different formulation will the conclusion change?

---

> ### Author Rebuttal · Authors · 2024-08-04
>
> Thanks for your comments. Please see the other responses for some clarifications to the mathematics that we will include in the final paper. We hope this will make the paper easier to follow.
>
> Regarding Lemma 7: it is indeed an important part of our analysis, and as we show it does indeed allow us to analyze Adam - in fact it arguably allows us to derive Adam from first principles. A similar argument using a less advanced online learning algorithm could be made to analyze SGD with momentum instead.

---

> > ### Comment · Reviewer_DXTx · 2024-08-12
> >
> > Thank you for the response I have raised my score.

---

### Official Review · Reviewer_BY48 · 2024-07-13

**Soundness:** 3
**Presentation:** 3
**Contribution:** 3
**Rating:** 7
**Confidence:** 3

**Summary:**

This paper proposes a variant of Adam involving per-iteration clipping of the updates (which look like Adam's updates) and EMA-based weight averaging. More specifically, two versions of this variant are proposed -- one is a global (w.r.t. all the coordinates) version and one is a per-coordinate version. The two main technical tools used in developing the proposed algorithm are (i) the online-to-nonconvex framework of references Cutkosky et al. [2023] and Zhang and Cutkosky [2024] wherein the update directions are chosen with an online learner, and (ii) choosing the online learner to be scale-free Follow-the-Regularized-Leader (FTRL) based on reference Ahn et al. [2024b]. The proposed algorithm attains the optimal convergence rate for both smooth and non-smooth nonconvex settings (although w.r.t. a different notion of stationary points for the non-smooth case). In addition, the benefit of the per-coordinate version is discussed.

**Strengths:**

**1.** The proposed algorithm is shown to attain the optimal convergence rate for both smooth and non-smooth non-convex problems.

**2.** I like the discussion on the benefit of coordinate-wise updates.

**3.** The results of this paper also shed some light on why EMA-based weight averaging can be useful in practice (although it doesn't seem strictly important in the context of the proposed algorithm; see Weaknesses).

**Weaknesses:**

**1.** It appears to me that this paper's main convergence results (Theorems 11 and 19) can hold even with $\mathbf{y}_t$ as defined in Lemma 7. Since Theorems 11 and 19 are *not* w.r.t. the last *averaged* iterate (which is what is used in practice), it seems that EMA-based averaging is not strictly necessary in the context of the proposed algorithms, i.e., the results hold even with the unaveraged iterates appropriately sampled (as described in Lemma 7).

**2.** I understand that this is a theoretical paper but because a new algorithm has been proposed involving clipping and EMA-based averaging, it would have been nice to show some empirical results at least in some simple settings. This would have made the proposed algorithm more appealing and convincing to practitioners.

**Questions:**

**1.** The purpose of Lemma 10 is not immediately clear to me. Is it to replace $\mathbf{y}_t$ in Lemma 7 with $\overline{\mathbf{w}}_t$?

**2.** It seems to me that clipping is always activated in $\beta$-FTRL and CLIPPED-ADAM(?) This is because with the choice of $\beta_1 = \beta$ and $\beta_2 = \beta^2$ in CLIPPED-ADAM for instance, $\sum_{s=1}^t \beta_1^{t-s} g_s \geq \sqrt{\sum_{s=1}^t \beta_2^{t-s} g_s^2}$.

**3.** Is the Lipschitz assumption necessary?

**Limitations:**

Discussed.

---

> ### Author Rebuttal · Authors · 2024-08-03
>
> Thank you for your constructive comments! Let us address your questions one by one.
>
> - (Weakness 1  / Question 1) We indeed need to choose $\mathbf{\bar{w}}_t = \mathbb{E}[\mathbf{y}_t]$ as the output, and this is crucial to achieve a  $(\lambda,\epsilon)$-stationarity, our main notion of convergence. This notion of convergence basically requires to output a point $\mathbf{w} = \mathbb{E}[\mathbf{y}]$ such that both $\mathbb{E}[\nabla F(\mathbf{y})]$ and $\mathbb{E}\|\mathbf{y}-\mathbf{w}\|^2$ are small. In other words, we need to find a point $\mathbf{w}$ for which **one can find a set of points around $\mathbf{w}$ whose averaged gradients is small**. In light of this, the role of the two lemmas are as follows:
>
>    - **Lemma 7** gives the set of points $\mathbf{y}_t$ that guarantees the smallness of $\mathbb{E}[\nabla F(\mathbf{y}_t)]$, and hence we need to output its expectation $\mathbf{\bar{w}}_t = \mathbb{E}[\mathbf{y}_t]$ as the output.
>   - **Lemma 10** ensures that $\mathbb{E}\|\mathbf{y}_t-\mathbf{\bar{w}}_t\|^2$ is small; in other words, it ensures that $\mathbf{y}_t$ is closely clustered around its average $\mathbf{\bar{w}}_t= \mathbb{E}[\mathbf{y}_t]$.
>
> - (Weakness 2) As discussed in the introduction section, Adam with EMA is very successful for large complex neural network models. Our main goal was to understand why such a method is very effective. We consider the clipping as a minor modification, and given that, our main message is justifying the success of Adam with EMA. Hence, we do not think running further experiments is necessary, because proposing a new algorithm is not the main scope of this work.
>
> - (Question 2) Actually, the inequality likely goes the other way so that clipping is a no-op. In practice, we expect the $g_s$ sequences to be random (due to stochastic noise, unstable training trajectories etc), for which the standard concentration inequalities yield $\sum_{s=1}^t \beta^{t-s}g_s \lesssim \sqrt{\sum_{s=1}^t (\beta^{t-s}g_s)^2}$. Hence, in practical settings, we expect the clipping to be unnecessary.
>
> - (Question 3) The Lipschitzness condition is indeed a standard assumption in the nonconvex nonsmooth settings. However, we note that one of the main messages of our main results is that Adam lets us adapt to the Lipschitzness constants (**coordinate-wise**) without the knowledge of them. Note that we almost certainly would require *some* kind of structural assumption on the "difficulty" of the loss function, so if we were to dispense with Lipschitzness it would likely be at the cost of some other assumption (such as smoothness).

---

> > ### Comment · Reviewer_BY48 · 2024-08-12
> >
> > Thanks for the rebuttal; I'll keep my score. Regarding the direction of the inequality, I believe my direction is correct (with exact inequality and not inequality up to constants) if all the $g_s$'s are non-negative (this can be seen by squaring both sides).

---

> > > ### Author Response · Authors · 2024-08-13
> > > **Thank you**
> > >
> > > You're right of course  that if all the  $\mathbf{g_t}$s have the same sign, then the clipping will occur. Our observation is that this is extremely unlikely to happen. The reason is the following: imagine that the algorithm is converging (even if it is converging somewhat slowly). In such a case, we would expect the gradients to be roughly mean-zero random variables (e.g. https://arxiv.org/pdf/2111.06328 or https://arxiv.org/pdf/1508.00882), in which case the clipping should be expected to not occur.
> > >
> > >   In fact, the case in which clipping does occur frequently results in "too good to be true" results that should provide further intuitive evidence of it being unlikely. For example, https://jmlr.org/papers/v18/17-079.html shows that if the prediction of FTRL is clipped on every round then in fact we would suffer only *logarithmic* regret rather than the standar $\sqrt{T}$ regret bound, and http://proceedings.mlr.press/v97/cutkosky19b.html shows that in the case that $\sum_{t=1}^T \mathbf{g_t}\ge \Omega(\sqrt{T})$ (even if clipping doesn't occur every single round), we can obtain preconditioning "for free" without any extra computation cost.

---

### Decision · Program_Chairs · 2024-09-25

**Decision:**

Accept (poster)

**Comment:**

This paper examines the benefit of model exponential moving average when the model is trained with Adam. In particular, the paper uses a discount-to-nonconvex conversion framework with a scale-free FTRL learner to obtain non convex optimization guarantees. This resultant algorithm resembles a clipped variant of Adam with model EMA, thus providing optimization guarantees for it. While there are differences with the version of Adam used in practice, the results are nevertheless insightful. The reviews for the paper were mostly positive except for one reviewer with low-confidence providing a lower score. While some of their concerns were valid, I believe the rebuttal mostly addresses them. I think this paper will be valuable addition to the conference. I recommend acceptance.